# Mapping as a probe for heating suppression in periodically driven quantum many-body systems

Etienne Wamba[1,2,3,4], Axel Pelster[1], James R. Anglin[1*]

**1** State Research Center OPTIMAS and Fachbereich Physik, Technische Universität Kaiserslautern, 67663 Kaiserslautern, Germany
**2** Faculty of Engineering and Technology, University of Buea, P.O. Box 63 Buea, Cameroon
**3** STIAS, Wallenberg Research Centre at Stellenbosch University, Stellenbosch 7600, South Africa
**4** International Center for Theoretical Physics, 34151 Trieste, Italy
* anglin@rhrk.uni-kl.de

August 17, 2021

## 1 Abstract

**Experiments on periodically driven quantum systems have effectively realized quasi-Hamiltonians, in the sense of Floquet theory, that are otherwise inaccessible in static condensed matter systems. Although the Floquet quasi-Hamiltonians are time-independent, however, these continuously driven systems can still suffer from heating due to a secular growth in the expectation value of the time-dependent physical Hamiltonian. Here we use an exact space-time mapping to construct a class of many-body systems with rapid periodic driving which we nonetheless prove to be completely free of heating, by mapping them exactly onto time-independent systems. The absence of heating despite the periodic driving occurs in these cases of harmonically trapped dilute Bose gas because the driving is a certain periodic but anharmonic modulation of the gas's two-body contact interaction, at a particular frequency. Although we prove that the absence of heating is exact within full quantum many-body theory, we then use mean-field theory to simulate 'Floquet heating spectroscopy' and compute the heating rate when the driving frequency is varied away from the critical value for zero heating. In both weakly and strongly non-linear regimes, the heating rate as a function of driving frequency appears to show a number of Fano resonances, suggesting that the exactly proven absence of heating at the critical frequency may be explained in terms of destructive interferences between excitation modes.**

## Contents

44 
45 

# 1   Introduction

47 If a quantum Hamiltonian depends on time periodically, then the system possesses a discrete
48 time translation symmetry, analogous to the discrete spatial translation symmetry of a lattice
49 potential. Analogous to Bloch waves, the periodically driven system allows a complete set
50 of solutions to the time-dependent Schrödinger equation, which have the form of a quasi-
51 energy phase factor times a time-periodic wave function. The Fourier series components of
52 the periodic wave function obey time-independent Schrödinger equations, and in this sense
53 periodic driving can effectively realize new time-independent Hamiltonians [1–3]. The growing
54 subject of *Floquet engineering* [4–10] seeks to exploit this possibility to simulate exotic many-
55 body dynamics that is not found in static condensed matter systems [11–13], in order to
56 answer fundamental questions or develop technological applications. Floquet engineering is
57 now a widespread tool in the realm of ultracold quantum gases [14–20]. Among many other
58 utilizations, it may allow achieving the Mott-insulator-to-superfluid transition of two-species
59 hardcore bosons [21].

60     The initial states that can actually be prepared in a driven system, however, may be limited
61 by the actual time-dependent Hamiltonian rather than by the corresponding Floquet effective
62 Hamiltonian, because it is still the time-dependent Hamiltonian which actually determines
63 the system's time evolution. Measurable observables likewise evolve under the actual time-
64 dependent Hamiltonian. Even though the Floquet effective Hamiltonian is time-independent,
65 therefore, it is a generic problem for Floquet engineering that a continuously driven system
66 typically suffers from heating [20, 22–26]. For initial quantum states that can be prepared in

experiments, the physical energy of the system, *i.e.* the expectation value of the actual time-dependent Hamiltonian, may not merely be periodic and bounded, but may grow secularly over long times. This long-term heating may mask the more interesting phenomena which are the target of Floquet engineering.

In a Fermi-Hubbard system [27], deviations from the expected behavior in the effective Hamiltonian may clearly arise for long modulation times when heating processes dominate. For quantum critical systems described at low energy by a conformal field theory, multiple dynamical regimes clearly occur depending on the drive frequency. For slow driving and long times, the system becomes unstable and heats up to infinite temperature. In the limit where the driving frequency is much faster than any natural frequencies of the problem, one can use the Magnus expansion to find an appropriate Floquet Hamiltonian which is robust against heating [3]. In some cases interference effects have also been shown to limit heating [25, 28]. Further examples showing ways to avoid heating by periodic driving are, however, of interest.

One technique, which has yielded exact results in many-body theory, is the use of space-time mappings to relate non-trivial systems to simpler ones. This approach has been applied to quantum gases in various special regimes of inter-particle interaction or dimensions [29–31]. In these cases the mappings have been performed by constructing exact non-trivial time-dependent many-body wave functions from simpler wave functions that were previously known, by appropriately transforming space and time coordinates. In general the previous use of space-time mappings in many-body theory has been restricted to looking for exact solutions in special cases. In a recent work, however, we showed that beyond the use of space-time mappings for exact solution, a much more general kind of exact mapping turns out to be possible between pairs of many-body time evolutions [32]. Even though the exact solutions may no longer be available for either evolution, the mapping between the two potentially very different evolutions remains exact. And in Ref [33], it was shown that that this intriguing mapping between many-body quantum systems can even be extended to open systems.

In the present paper, we apply our mapping to address the heating problem of Floquet engineering in quantum many-body systems. The structure of the paper is as follows: Section 2 reviews the quantum field mapping scheme for dilute quantum gases and uses it to construct a class of periodic modulations of the interaction strength, which must have exactly zero heating, because they can be mapped onto a time-independent system. Such a class is discussed in section 3 with a detailed example, where the evolution of a quantum gas with periodically driven interactions in a static harmonic trap is mapped onto the evolution of the gas with un-driven interaction in a (different) static harmonic trap.

Our main results begin in section 4, where we shift attention away from the exact mapping to investigate *why* heating vanishes in this special case, by examining a larger class of periodic modulations which includes our special zero-heating case, but should otherwise exhibit heating. In particular we consider a quasi-one-dimensional Bose gas in a harmonic trap, with contact interactions of periodically modulated strength; since exact solutions of the full quantum problem are unavailable, we fall back on Gross-Pitaevskii mean-field theory. Mean-field theory reproduces the exact quantum many-body result of zero heating at a critical modulation frequency (double the trap frequency), but also allows us to compute heating rates for a range of different driving frequencies, in different regimes of both driving amplitude and interaction strength. We find Fano-like resonances in the heating rate, suggesting a generic mechanism for heating suppression in Floquet systems.

In Section 5 we then apply our mapping to the cases with heating, mapping experiments

114 with modulated interactions in static traps onto experiments with constant interactions in
115 modulated traps. Here our results are more cautionary: plausible arguments based on the
116 modulated-trap version of the experiments may predict dramatic 'Bose fireworks' heating in
117 cases where it does *not* in fact occur, at least in mean-field theory. Section 6 concludes the
118 work and draws some perspectives.

## 119 2 Mapping and driving

120 In this section we review the exact mapping identities of [32] and [33] and use them to construct
121 a rapidly driven system which has zero heating because it is only a spacetime transformation
122 of an undriven system. Although the mapping is also applicable for general two-particle
123 interactions [32], we focus here on systems of dilute Bose gas with contact interactions.

### 124 2.1 Mapping identities

125 Consider a quantum gas in $D$ dimensions with particles of mass $M$ subject to contact two-
126 body interactions. Evolution of the gas is described in the Heisenberg picture of quantum
127 dynamics by a time-dependent quantum field operator $\hat{\psi}(\mathbf{r}, t)$ that satisfies the Heisenberg
128 equation of motion

$$i\hbar \frac{\partial}{\partial t} \hat{\psi} = -\frac{\hbar^2}{2M} \nabla^2 \hat{\psi} + V(\mathbf{r}, t)\hat{\psi} + g(t)\hat{\psi}^\dagger \hat{\psi}^2 , \tag{1}$$

129 where $V(\mathbf{r}, t)$ denotes the trapping potential, and $g(t)$ stands for the two-particle interaction
130 strength. Highly controllable time-dependent interactions are routinely achieved in current
131 quantum gas laboratories, for example via Feshbach resonance management [34–38]. Ex-
132 periments with time-dependent interactions are currently of high interest in investigating
133 non-equilibrium many-body evolutions [39–41].

134    Our spacetime mapping identities is the following. If $\hat{\psi}_A(\mathbf{r}, t)$ is a solution to (1) for
135 potential $V = V_A(\mathbf{r}, t)$ and interaction strength $g(t) = g_A(t)$, then the following $\hat{\psi}_B(\mathbf{r}, t)$ is a
136 solution for the following $V = V_B$ and $g = g_B$:

$$\hat{\psi}_B(\mathbf{r}, t) = e^{-\frac{iM}{2\hbar}\frac{\dot{\lambda}}{\lambda}r^2} \lambda^{D/2} \hat{\psi}_A(\lambda \mathbf{r}, \tau(t))$$

$$V_B(\mathbf{r}, t) = \lambda^2 V_A(\lambda \mathbf{r}, \tau(t)) + \frac{Mr^2}{2}\lambda^3 \left(\frac{1}{\lambda^2}\frac{d}{dt}\right)^2 \lambda$$

$$g_A(t) \mapsto g_B(t) = \lambda(t)^{2-D} g_A(\tau(t))$$

$$\tau(t) = \int_0^t \lambda(t')^2 dt' , \tag{2}$$

137 where $\lambda = \lambda(t)$ is an arbitrary function subject only to the constraints $\lambda(0) = 1$, $\dot{\lambda}(0) = 0$, for
138 $\dot{\lambda}(t) \equiv d\lambda/dt$. If we impose the Heisenberg-picture initial condition $\psi_A(\mathbf{r}, 0) = \psi_B(\mathbf{r}, 0)$ then
139 the two time-dependent field operators $\hat{\psi}_{A,B}$ describe two different experiments on a dilute
140 Bose gas prepared in the same initial state. Since $V_A$ and $V_B$ as well as $g_A$ and $g_B$ can easily
141 be quite different, the A and B experiments can involve very different manipulations of the
142 gas sample. Nonetheless the two second-quantized destruction fields are exactly related by
143 this simple mapping, which involves a time- and space-dependent phase factor and a rescaling
144 of space, and which relates the two experiments at different times, such that $t_A = \tau(t_B)$.

145   Any possible experimental observables can be represented as expectation values of $N$-point
146 functions of the second-quantized field operators,

$$F_{\text{ex}}(\mathbf{R}, \mathbf{R}', t) = \left\langle \prod_{j=1}^{N} \hat{\psi}_{\text{ex}}^{\dagger}(\mathbf{r}_j', t) \prod_{j=1}^{N} \hat{\psi}_{\text{ex}}(\mathbf{r}_j, t) \right\rangle, \tag{3}$$

147 where the subscript ex refers to any of the experiments A and B. The mapping between the
148 quantum fields relates the $N$-point functions to each other as follows:

$$F_B(\mathbf{R}, \mathbf{R}', t) = \lambda^{ND} e^{-\frac{iM}{2\hbar} \frac{\dot{\lambda}}{\lambda} \sum_{j=1}^{N} (r_j^2 - r_j'^2)} F_A(\lambda \mathbf{R}, \lambda \mathbf{R}', \tau(t)), \tag{4}$$

149 where $t \equiv t_B$, $\tau(t) \equiv t_A$. Thus the mapping truly implies that either of the two experiments
150 is a perfect analog simulation of the other one, with any measurements at any times in one
151 experiment corresponding, according to (4), to measurements at corresponding (different!)
152 times in the other experiment. The mapping identities for field operators, trapping potentials,
153 interaction strengths and $N$-point functions hold for any initial state of the system, pure or
154 mixed and no matter how far from equilibrium it is, as long as the initial state is the same in
155 both experiments A and B.

156   One practical application of the mapping, as indicated in [32], is to use it to simulate
157 a more difficult experiment B exactly by mapping to it from a technically more feasible
158 experiment A. An example given in [32] was a mapping between an A in which the harmonic
159 trap is simply turned off (a ballistic expansion experiment) and a B in which the contact
160 interaction strength is ramped to infinity. The mapping is valid, however, for *arbitrary* $\lambda(t)$.
161 With periodic $\lambda(t)$, therefore, one can effectively achieve more complex periodically driven
162 experiments, for instance with periodic modulation of the interaction strength, by performing
163 only simpler ones, in which for example only the trapping potential is varied.

164   In this paper we will begin with the most trivial limit of this application: the effective
165 realization of a periodically driven experiment B from a *time-independent* experiment A. The
166 point of this especially simple mapping is not just that a time-independent experiment is
167 easier than a time-dependent one: it is that in a time-independent experiment there can be
168 no secular heating, and so therefore any experiment which can be mapped exactly onto a
169 time-independent one according to (2) must also avoid secular heating, even if it includes
170 driving.

## 2.2   Driving without heating

172 Any particular mapping between two experiments A and B is defined by the arbitrary function
173 $\lambda(t)$ of Eqs. (2). A concrete example, which as we will see will map an undriven evolution in
174 A onto an experiment B with periodic driving in the contact interaction strength $g_B(t)$, is

$$\lambda(t) = \frac{1}{\sqrt{\frac{1-\gamma^2}{2} \cos(2\omega t) + \frac{1+\gamma^2}{2}}}, \tag{5}$$

175 where $\omega$ and $\gamma$ are arbitrary constants, taken as positive without loss of generality.
176   This $\lambda(t)$ is periodic in time, with $\lambda(n\pi/\omega) = 1$, $\dot{\lambda}(n\pi/\omega) = 0$ for all integer $n$. The
177 general mapping (2) thus implies that

$$\hat{\psi}_A\left(\mathbf{r}, \tau(\frac{n\pi}{\omega})\right) = \hat{\psi}_B\left(\mathbf{r}, \frac{n\pi}{\omega}\right), \tag{6}$$

so that all possible observables in experiment A at times $t_A = \tau(n\pi/\omega) = \gamma^{-1}n\pi/\omega$ will exactly coincide with those in experiment B at times $t_B = n\pi/\omega$. If there is no secular heating in experiment A, therefore, there cannot be any secular heating in experiment B.

To ensure that there is no secular heating in experiment A, we simply choose

$$V_A = \gamma^2 \frac{M\omega^2 |\mathbf{r}|^2}{2} \tag{7}$$

for the same arbitrary $\gamma$ and $\omega$ that appear in $\lambda(t)$, and select any *time-independent* contact interaction strength $g_A = g_0$. This makes the Hamiltonian for the gas in experiment A completely time-independent. The mapping (2), however, yields

$$V_B = \frac{M\omega^2 |\mathbf{r}|^2}{2} \tag{8}$$

$$g_B(t) = g_A \lambda(t)^{2-D} . \tag{9}$$

Experiment B thus also has a static harmonic trap with frequency $\omega_B = \omega$, generally different (since $\gamma$ can be anything) from the trap frequency $\omega_A = \gamma\omega$ in experiment A. In experiment B, however, the contact interaction strength $g_B(t)$ is time-dependent whenever $\gamma \neq 1$ and the effective dimensionality of the trapped gas is $D \neq 2$.

In particular $g_B(t)$ is anharmonically modulated (except for the degenerate case $\gamma = 1$) with frequency $2\omega = 2\omega_B$ and with an amplitude that depends on $\gamma$, as illustrated for the case $\gamma = 1.5$ in Fig. 1. For a quasi-1D Bose gas ($D = 1$) we have simply $g_B(t) = g_0\lambda(t)$ when $g_A = g_0$ is constant; the time average of the interaction strength felt by the atoms in experiment B is

$$\langle g_B \rangle = \frac{\omega_B}{\pi} \int_0^{\pi/\omega_B} g_A \lambda(t)\, dt = \frac{2}{\pi} g_A \mathrm{K}(1 - \gamma^2), \tag{10}$$

where K denotes the complete elliptic integral of the first kind. The interaction strength $g_B(t)$ oscillates in time around $\langle g_B \rangle$, as in Fig. 1(b). This specific time dependence of $g_B(t)$ is naturally an experimental challenge to realize precisely but the experimental technology to achieve it for trapped ultracold gases certainly exists.

In spite of this possibly (depending on $\gamma$) strong modulation of $g_B(t)$, however, the exact quantum field mapping of (2) ensures that all observables in experiments A and B are always related, at the different times $t_B = t$ and $t_A = \tau(t)$, by the simple scaling relation (4), which in particular reduces to identity after every driving period. If the shared initial state of the two experiments is time-independent in A, then the time-dependent state in B will simply oscillate forever periodically. Regardless of the initial state, the evolution in A will obviously conserve energy, and since the mapping between the two systems is periodic, there can never be any secular growth in the energy in B.

The particular form of $\lambda(t)$ chosen in (5) is a convenient example because according to (2) it yields a time-independent $V_B$, so that only the interaction is modulated in experiment B. With a generic periodic $\lambda(t)$ the time-independent experiment A would be mapped onto a class of B experiments with arbitrary periodic driving in $g_B(t)$, but with a simultaneous modulation of $V_B(t)$ that has to be synchronized non-trivially with $g_B(t)$, in accordance with (2). Our mapped B experiments with exactly no heating are thus always quite special cases of periodic driving; we continue with (5) and its static $V_B$ for the rest of this paper simply because the cases with time-dependent $V_B$ are equally special and more complicated to describe.

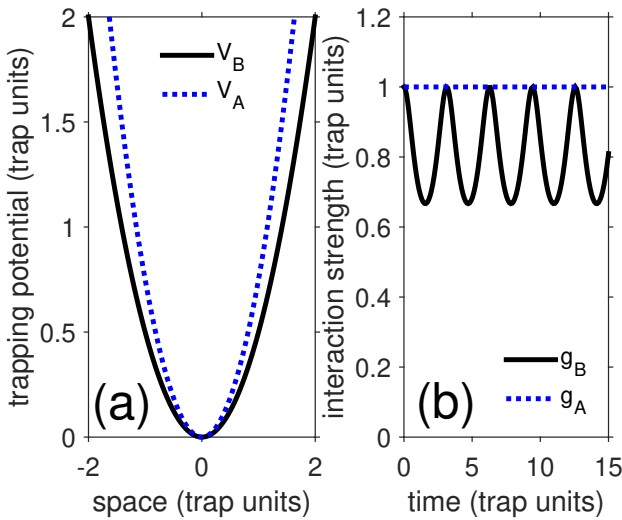

Figure 1: Sketch of (a) trapping potentials $V_B = M\omega_B^2 x^2/2$ and $V_A = M\gamma^2\omega_B^2 x^2/2$ over space $x$, and (b) interaction strengths in both experiments A and B over time $t$ according to Eq. (8). The time is measured in the trap units of Experiment B. Experiment A is a static evolution as both the trap and interaction strengths are kept constant in time. In experiment B, while the harmonic trap is static, the scattering length is periodically driven in time in such a way that the interaction strength is modulated with driving frequency $2\omega_B$. We used $\gamma = 1.5$ and $g_A = 1$, which yields $\langle g_B \rangle \approx 0.8$.

## 2.3 Mapping of times

We will describe our evolutions in the time $t = t_B$ of experiment B, but it is straightforward to derive the corresponding time in experiment A. From Eqs. (2), we find

$$t_A = \frac{\tan^{-1}[\gamma \tan(\omega_B t_B)] + n_B \pi}{\gamma \omega_B}, \tag{11}$$

where $n_B = \left\lfloor \frac{2\omega_B t_B + \pi}{2\pi} \right\rfloor$, with $\lfloor ... \rfloor$ denoting the floor function. Inversely, then we also have

$$t_B = \frac{\arctan[\gamma^{-1} \tan(\omega_A t_A)] + n_A \pi}{\gamma^{-1} \omega_A}, \tag{12}$$

where $n_A = \left\lfloor \frac{2\omega_A t_A + \pi}{2\pi} \right\rfloor$. Inserting (12) into (5), we can express the factor $\lambda(t)$ in terms of the time in experiment B as $\lambda(t(\tau)) =: \tilde{\lambda}(\tau) \equiv \tilde{\lambda}(t_A)$

$$\tilde{\lambda}(t_A) = \sqrt{\frac{1-\gamma^{-2}}{2}\cos(2\omega_A t_A) + \frac{1+\gamma^{-2}}{2}}. \tag{13}$$

The reciprocal relationship between $\lambda$ in (5) and $\tilde{\lambda}$ in (13) is generic for the spacetime mapping (2): the inverse mapping from B back to A is always simply the mapping with $\lambda \to 1/\lambda$ and $t_A$ and $t_B$ exchanged.

### 2.4  Why the absence of heating?

Our mapping has thus already shown the existence of a class of special cases of periodically driven quantum many-body systems with exactly no secular heating. Our further goal in this paper is to shed light on the mechanism by which these special cases avoid heating, since this mechanism will likely operate to some degree in a much broader range of cases of driving and is therefore of general interest. Since we cannot actually solve the full quantum many body problem, however, we will proceed to investigate dynamical mechanisms for avoidance of heating within Gross-Pitaevskii mean-field theory for the quasi-one-dimensional (quasi-)condensed Bose gas with weak contact interactions. It is straightforward to show [32] that the mapping (2), which is exact in the Heisenberg picture of the full quantum theory, is also valid in the corresponding mean-field theory.

## 3  Mapping and driving in mean-field theory

We illustrate concretely how the mapping between the time-independent and periodically driven experiments works within 1D mean-field theory for condensed bosons.

### 3.1  Numerical experiments with a pair of evolutions

We consider as a sample quantum gas a cigar-shaped (to the point of being quasi-one-dimensional) Bose-Einstein condensate that is described with a $c$-number field $\psi(x,t)$, the condensate wave function, governed by a Gross-Pitaevskii (GP) equation which is the mean-field counterpart of the Heisenberg equation (1),

$$i\hbar\frac{\partial}{\partial t}\psi = -\frac{\hbar^2}{2M}\frac{\partial^2}{\partial x^2}\psi + \frac{M[\omega(t)]^2 x^2}{2}\psi + g(t)|\psi|^2\psi \ . \tag{14}$$

The interparticle interaction is constant and the gas is confined in a harmonic trap with trap frequency $\omega(t)$ as sketched in Fig. 1(a). In order to solve the GP equation numerically, we prepare the initial state within the Thomas-Fermi regime using imaginary time relaxation. The same initial state will be used for both experiments A and B.

   In experiment A the un-driven system will simply remain in its initial ground state forever, while in experiment B the interaction strength is periodically modulated according to Eq. (8); as one could anticipate the gas density profile will not remain constant in B, since in B the Hamiltonian is periodically time-dependent. The implication of our mapping, however, is that all that will happen in B is a collective breathing mode, of which the amplitude will remain constant forever with zero secular growth. Without invoking our mapping, but simply numerically solving the Gross-Pitaevskii equation for B with the initial state as in A, we indeed obtain just such a breathing: see Fig. 2(b).

### 3.2  Mapping the two numerical experiments

We now directly confirm that experiments A and B as shown in Fig. 2 are mapped onto each other by (2) with the operator fields $\hat{\psi}_{A,B}$ replaced by the c-number order parameters $\psi_{A,B}$. Figure 3 shows the correspondingly obtained densities $|\psi_B(x,t)|^2_{\text{map}}$ and $|\psi_A(x,t)|^2_{\text{map}}$ obtained by mapping the densities $|\psi_A(x,t)|^2$ and $|\psi_B(x,t)|^2$ given by the Gross-Pitaevskii evolution that were displayed in panels (a) and (b) of Fig. 2. Comparing Figs. 2 and 3, it is

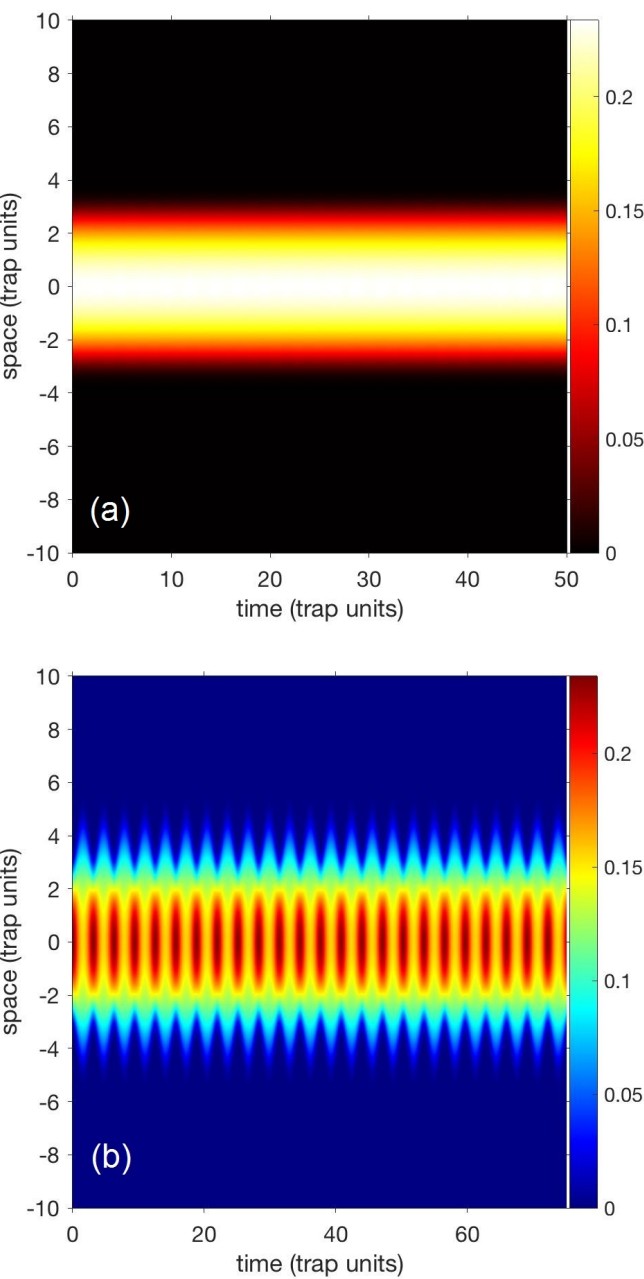

Figure 2: Density evolution in space and time $|\psi(x,t)|^2$ in two different experiments, namely (a) a static problem A and, (b) a Floquet problem B. In experiment A, we used the trapping frequency $\omega_A = 1.5\omega$ and the interaction strength $g_A = g_0 = 1.0$. In experiment B, the trapping frequency is $\omega_B = \omega$ and the interaction strength $g_B(t) = g_0\lambda(t)$ for $\lambda(t)$ given by (5) with $\gamma = 1.5$. The time is measured in units of $1/\omega$ and position in units of the corresponding trap length.

impossible to tell that the plots have not just been swapped for each other. The mapping is exact.

262    The mapping is however not trivial. The space and time axes in both plots, along with the
263 density scales, have been transformed according to (2) and its inverse. Note in particular the
264 difference of the time spans in panels (a) and (b). At the end of the displayed experiments,
265 we have $t_A \equiv 50$ and $t_B \equiv 75$ in the same trap-B-based natural units, as is readily obtained
266 from Eq. (11).

## 3.3   How is heating avoided?

268 The complete absence of heating in this special one-parameter family of periodic driving ex-
269 periments, with the interaction strength modulated at exactly twice the static trap frequency
270 and with a very particular $\gamma$-dependent anharmonic time dependence, makes this special case
271 interesting. It shows by example that secular heating can be avoided. In itself it is a mere
272 curiosity, though. The more generally interesting phenomenon which this special case may re-
273 veal is the dynamical mechanism of heating avoidance, since this mechanism can be expected
274 to operate, with greater or lesser effect, in a wide range of cases.
275    We therefore expand our attention now to a wider range of periodically driven experiments,
276 beyond those which lack heating because they map onto undriven experiments. In particular
277 we consider experiments of the same form as our previous experiment B for $D = 1$ as above,
278 and in which the temporal modulation of $g(t)$ has the same anharmonic form (5,8) as in our
279 mapped B experiments, but now with an arbitrary driving frequency:

$$g(t) = g_0 \left( \frac{1 + \gamma^2}{2} + \frac{1 - \gamma^2}{2} \cos(\nu\omega t) \right)^{-1/2} \tag{15}$$

280 for arbitrary real $\gamma$, $g_0$, and $\nu$. Our mapping results so far show that there will be no heating
281 for $\nu = 2$. What happens away from $\nu = 2$?

## 4   Heating rate and suppression

283 In this our main section we first face the basic question of how driving-induced heating can be
284 quantified from numerical computations within mean-field theory. We show how the heating
285 rate can be computed numerically. We then employ this method to see how heating rate in
286 our driven 1D (quasi-)condensate varies with driving frequency $\nu\omega$ and interaction strength $g$.
287 We will find that the heating rate shows troughs and peaks of a particular form that suggests,
288 by analogy with other dynamical systems, that heating avoidance occurs through destructive
289 interference of competing collective modes.

### 4.1   Numerical method for estimating heating

291 The secular heating rate is defined here as the average rate of change of the instantaneous
292 energy of the system at long times; we express the heating rate dimensionlessly in terms of
293 trap units $\hbar\omega^2$. In order to compute this heating rate, we first compute the instantaneous
294 energy and then determine its long-term average. As a trivial simplification we subtract the
295 initial energy and compute $\Delta E(t) = E(t) - E(0)$.
296    Within Gross-Pitaevskii mean-field theory for the (quasi-)condensed 1D Bose gas, the

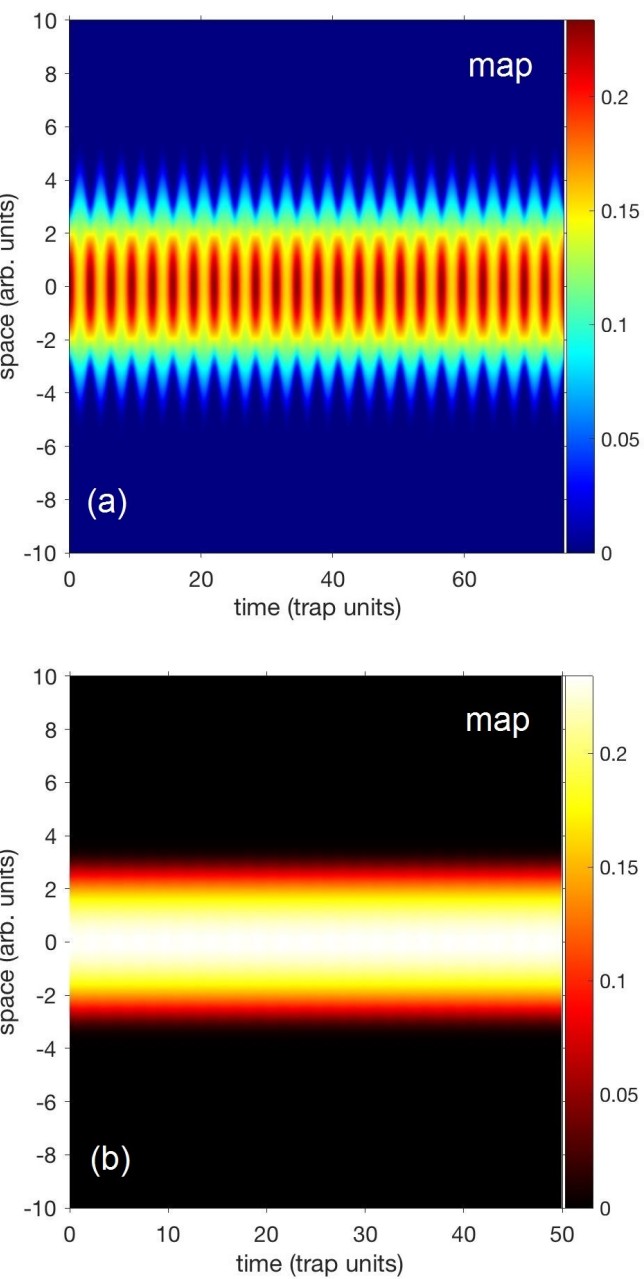

Figure 3: Space-time evolution of the densities (a) $|\psi_B(x,t)|^2_{\mathrm{map}}$ and (b) $|\psi_A(x,t)|^2_{\mathrm{map}}$, obtained by mapping the densities obtained in experiments A and B, respectively, using the relation (2) and its inverse. There are no differences between these plots and those of Fig. 2, confirming that our quantum field mapping is also exact in mean-field theory.

297     instantaneous energy is

$$E(t) \to E_{\mathrm{GP}}(t) = \int dx \left[ \frac{\hbar^2}{2M} \left| \frac{\partial \psi}{\partial x} \right|^2 + V(x)|\psi|^2 + \frac{g(t)}{2} |\psi|^4 \right]. \tag{16}$$

298  The energy difference $\Delta E(t)$ is in general not constant and may have complicated temporal
299  behavior. It exhibits multiple time scales, including a driving period and a beat period, as
300  well as some longer time scales. At large enough times, however, we find numerically that
301  $\Delta E(t)$ becomes dominated by a linear growth with a well-defined slope. We identify this slope
302  as the heating rate.

303      It is straightforward to detect the emergence of the linear energy growth because it con-
304  tinues steadily until it dominates clearly. We therefore simply evolve numerically under the
305  Gross-Pitaevskii nonlinear Schrödinger equation (14) over a total time $\tau$ (many trap periods),
306  recording the energy $\Delta E(t)$ at a discrete set of evenly spaced $t_n$ which are all whole-number
307  multiples of the driving period $2\pi/(\nu\omega)$. On this sequence of $\Delta E(t_n)$ we then perform a linear
308  regression analysis, fitting it to the linear model

$$\Delta E(t) \approx \mathcal{E} + \beta_\tau t, \quad \text{with } t \in [0, t_{\max}]. \tag{17}$$

309  See Fig. 4(a). While the intercept fitting parameter $\mathcal{E}$ is of no particular importance in our
310  problem, the slope or gradient fitting parameter $\beta_\tau$ represents the secular power gain rate of
311  the system, due to the driving, over the time scale $\tau$. By comparing $\beta_\tau$ for different large
312  values of $\tau$ (up to thousands of trap periods) we find that although there is an initial transient
313  regime in which $\beta_\tau$ varies significantly with $\tau$, at large enough $\tau$ the heating rate approaches
314  a constant (see (Fig. 4(b))), which we then identify as the heating rate $\Gamma := \beta_\infty$.

### 4.2  Heating rate for different interaction strengths

316  Thanks to sophisticated Feshbach techniques available in present-day quantum gas laborato-
317  ries, numerous experiments have been achieved with variable interaction strengths [34–38].
318  We therefore pause briefly here to investigate how the heating of our 1D mean-field gas is
319  affected by the interaction strength prefactor $g_0$. This serves as a generic check on our method
320  for determining heating; we must expect that heating is generally weaker for weakly inter-
321  acting systems that are periodically driven, since driven harmonic oscillators reach constant-
322  amplitude steady states, except exactly on resonance.

323      Since the only driving in our system is in the interaction we can have no heating at all for
324  $g_0 \to 0$, but we can confirm the reasonable behavior of our numerical heating rates by seeing
325  how they tend to increase with $g_0$, as shown in Fig. 5. Globally, the heating rate increases
326  with the interaction strength for all driving frequencies $\nu\omega$. The average rate of heating rate
327  increase with $g_0$ itself increases with the driving frequency. In agreement with the results in
328  our earlier sections, the heating rate remains zero for all $g_0$ in the special non-heating case
329  $\nu = 2$ that can be mapped onto the time-independent system by the space-time rescaling (2).
330

### 4.3  Heating rate for different driving frequencies: heating avoidance and hidden adiabaticity

333  We now proceed to consider the effect on heating of the driving frequency $\nu\omega$; this is the core of
334  our paper, and the heating spectrum shown in Fig. 6 is our main numerical result. Our results
335  shown for $t_{\max} = 2500$ are not discernibly different from those with $t_{\max} = 1125$, confirming
336  that we are analyzing the asymptotic long-time regime of secular heating. For $\nu = 0$ heating
337  must vanish exactly, since the system is static, and for sufficiently low $\nu$ the system should still
338  avoid heating, because it should adapt to the slowly modulated Hamiltonian adiabatically.

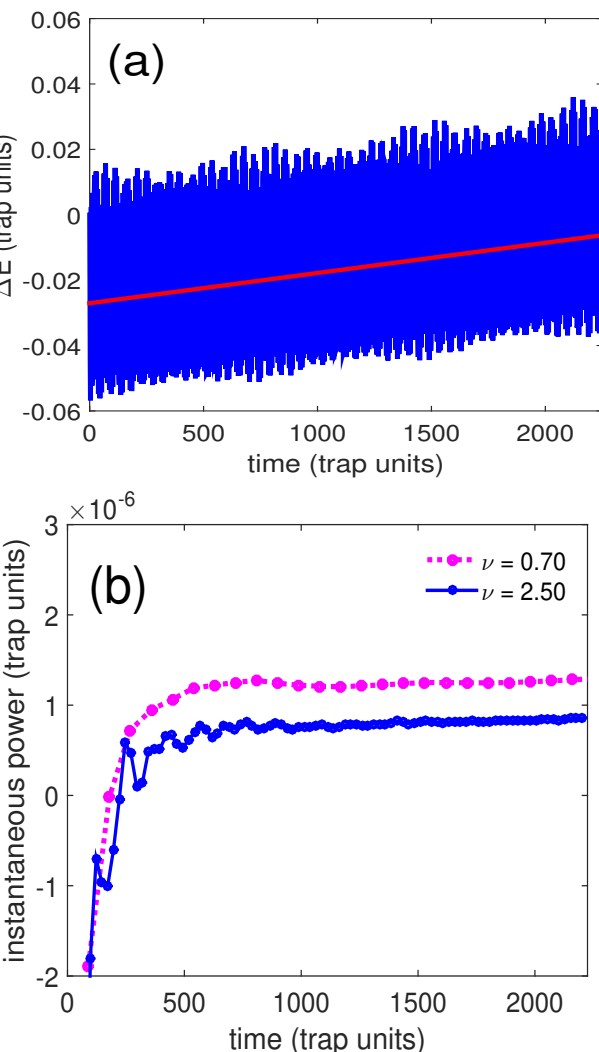

Figure 4: (a) Time evolution of the energy difference $\Delta E$ along with the fitting model (red straight line). (b) Instantaneous power $\beta_\tau$ received by the gas from the drive for two different driving frequencies $\nu = 0.7$ and 2.5; we used the interaction strength in the form (8) with $g_0 = 1.0$ and modulation amplitude parameter $\gamma = 1.5$. The power is measured in trap units of $\hbar\omega^2$. The time when the power stops changing considerably allows us to determine the long-time regime, which is roughly $t \in [600, \infty[$ and $[1000, \infty[$ for the frequencies $\nu = 0.7$ and 2.5, respectively. To obtain the power $\beta_\tau$, we considered $\Delta E(t)$ at times $t = 2(n+n_0)\pi/(\nu\omega) \equiv t_n$, for $n = 0, 1, \cdots, N$ and $t_N \leq t_{\max}$. Then we obtained $\beta_\tau$ through a linear fit of the data in the set $\{\Delta E(t_0), \cdots, \Delta E(\tau)\}$ for $\tau \in \{t_1, \cdots, t_N\}$. We used $n_0 = 4$.

We do not see this in Fig. 6, however, because in fact the lowest $\nu$ that we computed was $\nu = 0.05$, which is evidently not slow enough for the system to remain adiabatic over very long times. For $\nu < 2$ we do see the heating increasing slightly with drive frequency, as we noted in Fig. 5 above. Over the extended range $2 < \nu < 10$, however, this increasing trend does not continue and instead the background heating rate remains nearly constant. Heating should again decrease trivially at very high frequencies, as the system responds only to the

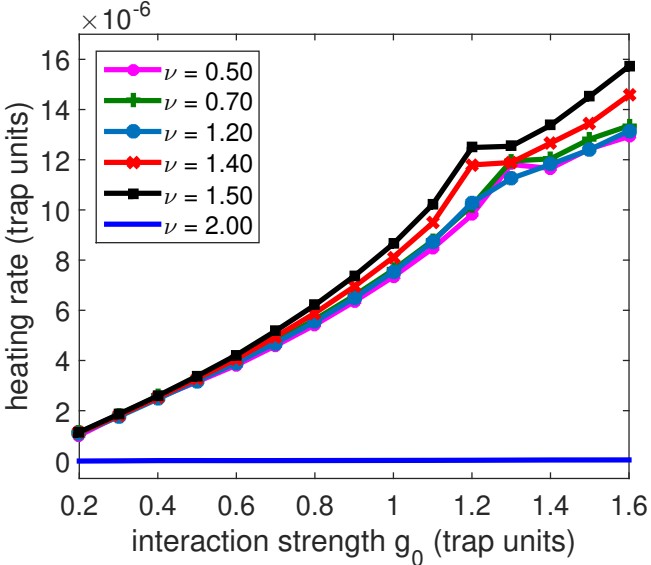

Figure 5: Heating rate as a function of interaction strength at small driving frequencies and at the heating trough frequency. The frequencies are given by $\nu = 0.5$, 0.7, 1.20, 1.40, 1.5 (less than the sharp resonance frequency $\nu \approx 1.9$) and 2.0 (heating zero); we used the runtime $t_{\max} = 1125$ and driving strength parameter $\gamma = 1.5$.

static time-averaged potential, but our one-dimensional Gross-Pitaevskii system has many high-frequency collective modes, and it is a nonlinear system with finite-amplitude driving; the time-averaged high-frequency limit is clearly well above $\nu = 10$.

Against the essentially flat background heating in Fig. 6, two dramatic features are seen: sharp dips and spikes in the heating rate at a number of particular driving frequencies.

**Heating spikes.** Many heating spikes can be seen in the heating spectrum. They appear close to frequencies that are associated with parametric resonances [42–45]. In general parametric resonance can be excited whenever the driving frequency $\nu\omega = 2\Omega_n/m$, where $\Omega_n$ is an eigenfrequency of the system and $m$ any positive integer.

For vanishing interactions the eigenfrequencies in our dimensionless trap units are $\Omega_n = n\omega$, with $n$ being any positive integer. With $g_0 = 1$ the collective mode frequencies as given by Bogoliubov-de Gennes linearization of the Gross-Pitaevskii equation (14) around our initial ground state are slightly shifted from the non-interacting frequencies; they are shown as dotted vertical lines in Fig. 6. In the Thomas-Fermi limit where there is an infinitely large interaction strength, we anticipate that the heating spikes would appear close to the eigenfrequencies $\Omega_n^{\mathrm{TF}} = \sqrt{n(n+1)/2}\,\omega$. It is clear that most of our heating spikes are appearing very close to these collective mode resonances. The reflection symmetry of our trapping potential and initial state means that only even-parity collective modes can be excited by our driving.

Heating spikes at odd-parity Bogoliubov-de Gennes frequencies, or in between frequencies, are due to subharmonic excitation with $m > 1$. With $\gamma = 1.5$ our periodic driving is significantly but not extremely anharmonic, so that subharmonics with large $m$ or of higher modes do not seem to cause significant heating, but subharmonics with $m = 2$ or $m = 3$ are clearly visible for some of the lower modes, as seen in Fig. 6(b). The moderate driving amplitude of

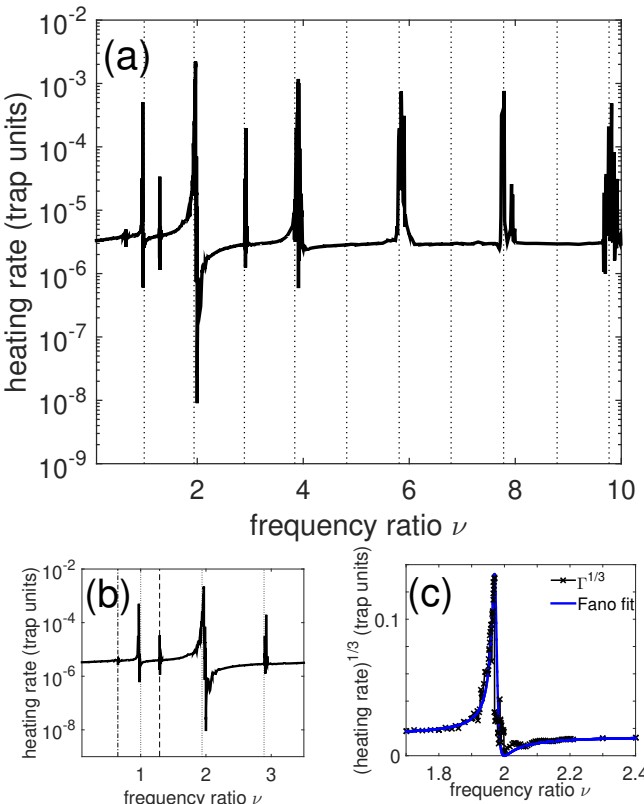

Figure 6: Heating rate as a function of driving frequency $\nu\omega$ for runtimes taken in the long-time regime; we used the interaction strength $g_0 = 1.0$ and driving strength parameter $\gamma = 1.5$. The horizontal axis does not really begin at $\nu = 0$, but at $\nu = 0.05$ (the lowest value we simulated), and so the quasi-static limit of very small $\nu$ does not appear in the plot. Dotted lines correspond to Bogoliubov – de Gennes excitation frequencies while dash-dotted and dashed lines correspond to the $1/3$ and $1/2$ of these frequencies, respectively. (a) Heating rate in the frequency spectrum in the semilog plot; (b) A zoom of the range of smaller frequencies where subharmonics are excited; (c) The heating rate to the power $1/3$ is fitted to a Fano resonance peak around the exact heating zero at $\nu = 2$.

368  $\gamma = 1.5$ is also evidently sufficient to produce slight nonlinear shifting of the resonance peaks
369  away from the linear Bogoliubov-de Gennes resonances. We realize that the heating spikes
370  lower and broaden as the frequency is getting higher. In addition the heating rate slowly
371  softens on average as one would naturally expect when the frequency increases.

372  **Heating trough.**  As noted above, there must be a trivial heating minimum around $\omega = 0$
373  , because for very slow modulation of $g$ the system will react adiabatically; this trough is not
374  seen in Fig. 6 because we do not actually show results below $\nu = 0.05$. Our focus is not on
375  the adiabatic limit, but rather on the non-trivial heating minima such as the one at $\nu = 2$,
376  which is the zero-heating case that was identified above by our mapping. From the zoomed-in
377  Fig. 6c) it is clear that this zero-heating case is not a unique point, but rather the bottom of
378  a finite heating trough of low heating rates.

379 **Fano resonances.** Very close to this heating trough, furthermore, there is a huge heating
380 spike which may suggest the existence of a hidden compensation mechanism leading to the
381 heating suppression. In the low heating region, the system evidently responds to the drive
382 in a nearly adiabatic way. Even though adiabatic following is normally observed only for
383 slow external driving, the heating trough thus reveals a kind of hidden adiabaticity in a
384 rapidly driven many-body system. The appearance of the heating trough close to a heating
385 spike, forming a distinctly asymmetric trough-peak pattern, reminds us of the so-called *Fano*
386 *resonance* that occurs in nanoscale structures [46,47]. Even though very mild oscillations of
387 the actual heating curve $\Gamma(\nu)$ can be seen during the sharp resonance and heating trough, the
388 curve is fit quite well by the Fano function, which in this case is given by

$$\Gamma^{1/3} = 1.55\,\sigma\frac{(\nu-2)^2}{\sigma^2 + (\nu-2-\delta)^2}, \tag{18}$$

389 where the dimensionless width is $\sigma = 1/105$ and the asymmetry is due to $\delta = -0.027$. This
390 Fano profile is shown in Fig. 6(c). Fano resonances are the result of an interference between
391 an excitation of a single mode and an excitation of a broad spectrum of modes [47]. Such a
392 process occurs for example in atomic physics, in the excitation of an electronic configuration
393 that has an energy higher than that needed to ionize the atom. Fano resonances generally
394 appear in the context of single-particle systems where Floquet theory is applicable to the linear
395 equations that describe the system [50,51]. In those systems, Fano resonances at dynamically
396 created bound states in the continuum may lead to points of zero transmission where the
397 so-called quantum resonance catastrophe occurs. In this work, however, similar resonances
398 happen in the realm of many-body physics where interactions between the atoms are normally
399 expected to yield a more complex behavior. A deeper investigation of such resonances would
400 require elaborate methods that are beyond the scope of this paper but may be addressed in
401 future work; here we simply observe that our numerical mean-field results seem to suggest
402 their existence in periodically driven interacting Bose gases.

## 403 5 Mapping interaction modulation to trap modulation

404 The 'heating rate spectroscopy' of section 4 has identified Fano-like resonances in the heating
405 rate as a function of the frequency with which the contact interaction strength of a dilute
406 Bose gas is modulated, while the gas remains trapped in static harmonic potential of trap
407 frequency $\omega$. The resonances appear near collective mode frequencies and their subharmonics;
408 close beside these resonance peaks are narrow minima (troughs) in the heating rate. At the
409 special heating trough at drive frequency $\omega_{\text{drive}}/\omega = \nu = 2$, the heating rate falls all the way to
410 zero, as the exact mapping described in section 3 demands. While it is only this single special
411 heating trough that is mapped exactly onto a static experiment, we can still ask whether
412 the mapping may shed further light on heating in many-body Floquet systems, by mapping
413 experiments with $\nu \neq 2$ onto other experiments which might be easier to analyze.

### 414 5.1 Mapping to experiments with constant interaction strength

415 We therefore now consider the evolutions of section 4 as A experiments, with constant trap
416 frequency $\omega_A = \omega$ and modulated interaction strength $g_A(t) = g(t)$ as given by (15), with

417 arbitrary overall interaction strength $g_0$, modulation amplitude $\gamma$, and drive frequency $\nu\omega$.
418 The exact mapping (2) with

$$\lambda(t) = \sqrt{\frac{1-\gamma^2}{2}\cos(\nu\omega\tau(t)) + \frac{1+\gamma^2}{2}} \implies \tan\left(\frac{\gamma\nu\omega}{2}t\right) = \gamma\tan\left(\frac{\nu\omega}{2}\tau\right) \tag{19}$$

419 then yields a B experiment for every value of $\nu$ in which the interaction strength has been
420 mapped to the *time-independent* $g_B = g_A(\tau)/\lambda(t) \equiv g_0$ but, except for $\nu = 2$, the trap
421 frequency $\omega_B(t)$ is now a periodic function:

$$\omega_B^2(t) = \omega^2\lambda^4 + \lambda^3\frac{d^2\lambda}{d\tau^2} = \omega^2\left(\frac{1-\frac{\nu^2}{4}}{\left(\frac{1+\gamma^{-2}}{2} + \frac{1-\gamma^{-2}}{2}\cos(\gamma\nu\omega t)\right)^2} + \frac{\nu^2\gamma^2}{4}\right). \tag{20}$$

422 In the case $\nu = 2$ we thus recover the entirely static experiment with trap frequency $\gamma\omega$,
423 from which we constructed the heating-free driven experiment in the first place, via the inverse
424 of this mapping. For $\nu \neq 2$ we now have a class of experiments with constant interactions and
425 modulated trapping frequencies, which are mapped exactly as quantum many-body problems
426 onto the experiments that we analyzed in mean-field theory in section 4. We still cannot solve
427 these new evolution problems exactly, and solving them in mean-field theory will only yield the
428 image under the mapping of our results from section 4 above. It might be possible, however,
429 to obtain some insight or intuition about our heating peaks and troughs by considering these
430 physically quite different experiments which are exactly mapped versions of the previous ones.

## 5.2    Mapping the mean-field energy

432 We can establish that a B experiment will show long-term heating if and only if the cor-
433 responding A experiment shows long-term heating. This is intuitive but not quite obvious,
434 because it is straightforward to show that the mapping (2) transforms the Gross-Pitaevskii
435 mean-field energy (16) as

$$E_B(t_B) = \lambda^2 E_A(t_A) + \frac{M}{4}\left(\frac{d^2\lambda^2}{dt_A^2}\right)\int dx\, x^2|\psi_A(x,t_A)|^2$$

$$-\frac{M}{4}\left(\frac{d\lambda^2}{dt_A}\right)\frac{d}{dt_A}\int dx\, x^2|\psi_A(x,t_A)|^2 \tag{21}$$

436 for $t_A = \tau(t_B)$. Thus the energies in the two experiments are not simply the same.
437 Suppose, however, that $E_A$ does not show secular growth. In this case $\int dx\, x^2|\psi_A|^2$ cannot
438 show secular growth, either, because $E_A$ is a sum of positive definite terms, one of which
439 is proportional to $\int dx\, x^2|\psi_A|^2$, so that if $\int dx\, x^2|\psi_A|^2$ grew secularly then $E_A$ would have
440 to grow secularly as well. Thus if $E_A$ does not show secular growth, then neither can $E_B$,
441 because it consists only of non-growing terms multiplied by periodic functions.
442 Suppose now that $E_B$ does not show secular growth. As we observed above, it is straight-
443 forward to confirm from our mapping definition (2) that the inverse transformation which
444 maps from B to A is simply the mapping with $\lambda \to 1/\lambda$. Hence we also have

$$E_A(t_A) = \lambda^{-2}E_B(t_B) + \frac{M}{4}\left(\frac{d^2\lambda^{-2}}{dt_B^2}\right)\int dx\, x^2|\psi_B(x,t_B)|^2$$

$$-\frac{M}{4}\left(\frac{d\lambda^{-2}}{dt_B}\right)\frac{d}{dt_B}\int dx\, x^2|\psi_B(x,t_B)|^2. \tag{22}$$

445 Hence by the same argument that we have just made above, if $E_B$ does not show secular
446 growth then neither can $E_A$. Thus it is impossible for $E_A$ to grow secularly without $E_B$ also
447 growing secularly.

448      The questions of secular heating in A and B experiments are therefore really both the
449 same question. Whatever mechanisms cause or suppress heating in one kind of experiment
450 will be the images, under our mapping, of the mechanisms that cause or suppress heating in
451 the other kind of experiment. Unfortunately, however, this does not necessarily mean that
452 the mechanisms are obvious in either case. To illustrate the kind of subtle problem that can
453 occur even in the comparatively simple B experiments, in which only the trap potential is
454 modulated, we will propose an argument for explosive heating in some A experiments, which
455 should seem plausible but has in fact already been disproven by our results in section 4 above.

### 456    5.3    Absence of Bose fireworks from intermittent anti-trapping

457 **Repulsive potentials.** For $\gamma \neq 1$ and $\nu$ above a $\gamma$-dependent threshold $\nu_-(\gamma)$, $\omega_B^2(t)$ as
458 given by (20) can become negative within certain time intervals. The threshold $\nu$ above which
459 $\omega_B^2 < 0$ is

$$\nu > \nu_-(\gamma) = \begin{cases} \frac{1}{\sqrt{1-\gamma^2}} & , \quad \gamma < 1 \\ \frac{\gamma}{\sqrt{\gamma^2-1}} & , \quad \gamma > 1 \,. \end{cases} \tag{23}$$

460 There are no values of $\gamma$ and $\nu$ for which the trapping strength $\omega_B^2$ becomes negative for all $t$,
461 but for $\nu$ above a higher threshold $\nu_c(\gamma)$ the *average* trap strength over a driving period does
462 become negative:

$$\langle \omega_B^2 \rangle = \frac{\gamma\omega\nu}{\pi} \oint \omega_B^2(\tau)d\tau = \frac{\gamma\omega^2}{2}\left(1 + \gamma^2 - \frac{\nu^2}{4}(1-\gamma)^2\right)$$

$$\implies \nu_c(\gamma) = 2\frac{\sqrt{1+\gamma^2}}{|1-\gamma|} \equiv \nu_c(1/\gamma) \,. \tag{24}$$

463 **Bose fireworks?** It seems plausible that heating should increase in some significant way
464 when $\nu > \nu_-(\gamma)$, since then the quasi-one-dimensional gas is being repeatedly subjected to
465 a repulsive potential instead of a trap. And it seems plausible that heating should become
466 quite strong indeed for $\nu > \nu_c(\gamma)$, since then the gas is actually being anti-trapped, rather
467 than trapped, for most of the time. Moreover the thresholds $\nu > \nu_-$ for $\omega_B^2(t) < 0$ and $\nu > \nu_c$
468 for $\langle \omega_B^2 \rangle < 0$ have nothing to do with the mean-field approximation; they are facts about the
469 time-dependent potential strength $\omega_B^2(t)$ which remain true in the full quantum many-body
470 problem. It may therefore not seem too much to expect that our mapped B experiments are
471 here predicting something like the so-called 'Bose fireworks' that have been seen in A-like
472 experiments with modulated interaction strength [48, 49].

473 **No.** In fact, however, Fig. 6 has already shown that there is no substantial increase in
474 the heating rate for either $\nu > \nu_-$ or $\nu > \nu_c$. Whether or not this is counter-intuitive, the
475 particular form of modulated potential (20) simply does not cause any fireworks-like heating.
476 The slow, linear heating that we have seen in section 4 above does occur, as long as $g_0 \neq 0$,
477 $\gamma \neq 1$, and $\nu \neq 2$. In the B experiments this heating is produced by the modulated trapping
478 potential alone, with the interaction strength $g_B = g_0$ constant. The fact that this form

of potential modulation does not generate abundant heat by itself, however, can be seen by considering the non-interacting case $g_0 = 0$. One might expect the modulating potential to heat the non-interacting gas, but for $g_0 = 0$ the A experiment to which all the B experiments can be mapped has time-independent trap strength $\omega^2$ and also $g_A = 0$. There is therefore no heating at all for $g_0 = 0$ in either A or B experiments, for any values of $\nu$ or $\gamma$.

We must therefore recognize that neither $\omega_B^2(t) < 0$ nor even $\langle \omega_B^2 \rangle < 0$ has to imply dramatic heating. Evidently in the intervals of positive $\omega_B^2$ the trap can be strong enough to pull the gas back together again after it has been dispersed during the anti-trapping intervals of negative $\omega_B^2$.

The analogy between our A experiments with modulated interaction and the actual 'Bose fireworks' experiments is also evidently not as close as it might at first seem. First of all in [48, 49] the interaction strength $g(t)$ was much more strongly modulated than our $g(t)$ from (15) can allow for any $\gamma$: the real experiments had $g(t)$ oscillating between positive and negative values, with an amplitude some twelve times greater than the mean value of $g(t)$. With large $\gamma$ we can achieve arbitrarily large amplitude in $g(t)$ but our (15) does not allow $g(t)$ to change sign. Secondly the real experiments used a trapping potential of finite depth, and for highly excited atoms this is qualitatively different from our parabolic potential extending to infinity.

# 6  Conclusion and outlook

Periodically driven many-body quantum systems are a useful experimental tool for understanding non-equilibrium physics, but one phenomenon of non-equilibrium physics which they cannot in general avoid is the problematic phenomenon of secular heating. Our exact spacetime mapping between quantum fields provides a limited range of results for such problems, but in the special cases that the mapping provides, it is exact, and so it can supply us with instructive examples. We have used it to identify a special form of periodic modulation of the strength of the contact interaction in a dilute Bose gas: a special case in which there is no heating at all. This shows the possibility of a kind of hidden adiabaticity in rapidly driven interacting quantum systems.

We have further explored this phenomenon with numerical calculations for quasi-one-dimensional Bose-Einstein condensates in Gross-Pitaevskii mean-field theory, showing that the heating rate in this kind of system can show Fano-like resonances as a function of driving frequency. The exact zero-heating case found by our mapping appears to be one of these resonances, but many similarly narrow, deep troughs in the heating rate can also appear. From our failed speculations about dramatic heating in section 5, the cautionary lesson must be drawn that the exact mapping may not always be able to simplify complex experiments by mapping them onto simple ones. Sometimes it will instead reveal that the seemingly simple experiments are not really as simple as they seemed. This is also learning something, however. Further applications of the exact spacetime mapping of quantum fields [32,33] to periodically driven quantum many-body systems will be well worth pursuing.

## Acknowledgements

The authors would like to thank Sebastian Eggert and Christoph Dauer for discussions.

**Funding information**   EW acknowledges financial support from the Abdus Salam International Center for Theoretical Physics, through a Simons Associateship, and from the Alexander von Humboldt Foundation under the grant number 3.4-KAM/1159208 STP. The work is also funded by the Deutsche Forschungsgemeinschaft (DFG, German Research Foundation) under the project number 277625399 - TRR 185.

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
