# Peer review of "Mapping as a probe for heating suppression in periodically driven quantum many-body systems"

_SciPost Physics_

## Round 1 · Referee Report · Thomas Bilitewski (Referee 1) · 2021-10-2

Strengths

1- establishes an exact mapping proving absence of heating in a class of interacting periodically driven quantum systems

2- establishes (numerically) the suppression of heating not only at the exact mapping point, but for deviations in the driving, and at a number of resonance far from this point

Weaknesses

1- While the mapping is exact it does applies to a rather special class of models only. It is not immediately obvious that many of the interesting interacting many-body phases of interest in Floquet engineering could benefit/be realised within the constraints of being able to be mapped to a static system.

2- While the authors try to connect the suppression of heating observed in the exact mapping to wider classes of periodically driven systems, and interpret the resonances found as "Fano" resonances potentially active more generally, the evidence for this seems to be mainly a good fit to the numerically observed resonance features.

Report

The manuscript "Mapping as a probe for heating suppression in periodically driven quantum many-body systems" discusses the use of space-time mapping identities applied to the case of harmonically trapped contact interacting Bose gases to establish the absence of secular long-time heating for a specific class of periodic modulation of system parameters.

In addition to discussing the exact mapping, the authors further establish the suppression of heating close to the exact mapping point, and at a number of further resonances they interpret as Fano resonances in numerical simulations of the mean field GP equation.

The field of Floquet engineering of interacting many-body phases is highly-active, and exact results on the absence of heating in interacting systems are certainly of great interest. The work appears perfectly technically sound, and seems to indicate some applicability beyond the exactly mappable cases.

However, I am not fully convinced that the results established here will easily translate to more general situations relevant for the realisation of non-trivial phases. In addition, while the application to periodically driven situations is novel, the mappings themselves have been established by the authors in prior work (Ref [32,33]).

The work satisfies all general acceptance criteria, I am not fully certain if it satisfies the expectation criteria. Given that the general mapping has been identified before I don't see it as groundbreaking or a breakthrough. I believe the strongest argument could be made for it opening up a new way of studying periodically driven systems, with some potential for future work, for which I am willing to recommend publication.

Requested changes

1- I believe a more thorough discussion of avoiding heating in Floquet systems in the introduction might be beneficial. The introduction seems to skip over the field of pre-thermalisation, including the rigorous bounds on heating rates, and the field of MBL and disorder which are two of the dominant ways to engineer interacting non-trivial Floquet phases.

2- The manuscript uses "trap-units" /"trap length" for space, time and energy at various points, e.g. in all figures. I couldn't find an explicit definition of those, and if those refer to trap parameters for experiment A or B separately. A definition before they are first used might be useful, in particular since for time-dependent trap parameters it's not immediately obvious what those would be.

3- The GP equation (Eq 14) as a non-linear equation requires a normalisation for $\int dx |\psi|^2$. Since there are two common conventions of either $\int dx |\psi|^2 = N$, or $\int dx |\psi|^2 = 1$ absorbing the N factor into the definition of g, the authors should specify which convention they are using.

4- I'd like some clarification on a small point in the argument in section 5.2 on equivalence of long-term heating between experiment A and B. The authors discuss non-growth of $\int dx |\psi_A|^2$, and claim this implies non-growth of Eq.(21). That doesn't seem to follow without additional assumptions on the time-dependence since $\frac{d}{dt_A} \int dx |\psi_A(x,t_A)|^2$ can grow without bound even if $\int dx |\psi_A|^2$ is bounded.

I assume the situation of periodically-driven systems might impose additional constraints that make the statement correct, in which case that might be worth clarifying.

Minor changes: 5- The use of "quasi-Hamiltonian" in the abstract seems uncommon to me. Most of the literature seems to rather use effective Hamiltonian

6- Eq (11) uses $\tan^{-1}$, and Eq (12) uses $\arctan$

7- Caption of Fig 5 refers to "less than the sharp resonance frequency $\nu \approx 1.9$". I assume the authors refer to the resonance found in Fig. 6. However at this point in the text this resonance hasn't been mentioned, and would need some context.

---

## Round 1 · Referee Report · Anonymous (Referee 2) · 2021-10-12

Strengths

1- Self-contained and clear presentation of the research background and new results 2- Interesting idea of connecting time-dependent and -independent problems 3- Complete numerical and theoretical investigation of a specific problem

Weaknesses

1- The studied model seems to be quite special and to have a limited range of applicability. 2- Lack of implications for Floquet engineering and Floquet heating in generic systems

Report

In this paper, the authors theoretically study the heating problem in periodically driven quantum many-body systems. The authors devise a special class of driven Hamiltonians using the spacetime mapping technique that the authors developed in their previous studies. Under these driven Hamiltonians, the quantum system does not heat up, unlike under generic Hamiltonians. The authors discuss how to construct such Hamiltonians and address why the heating does not happen. These Hamiltonians never lead to heating because they are, by construction, equivalent to a time-independent Hamiltonian.

Although the idea is interesting and the analysis is well-done, I am worried about how this study contributes to understanding Floquet engineering or Floquet heating, which are emphasized to motivate their research in the Introduction.

(1) Floquet engineering: If an experimentalist realizes a (spacetime-mapped) time-dependent Hamiltonian, is it useful? I think this is the Floquet engineer's viewpoint, and no heating is not enough.

(2) Floquet heating: I am not convinced that the heating studied in this work is related to Floquet heating in the general context. As is well known (e.g., in Ref. [3]), the Floquet heating is characteristic to periodically driven *many-body* Hamiltonians. However, the authors use the mean-field approximation, which approximates the original many-body Hamiltonian to a one-body one. Is it possible to talk about the Floquet heating within the one-body approximation? I am skeptical about this because I do not see the prominent exponentially slow heating [Phys. Rev. Lett. 115, 256803] in authors' Figs. 5 and 6.

Unless either (1) or (2) is resolved, the implications of the authors' results would be quite limited. Of course, I am sure that they scrutinize the specific model very carefully and professionally.

Considering the high-standard acceptance criteria of SciPost Physics, I find neither of the four expectations met while general acceptance criteria are met. Thus, I recommend that the authors send their manuscript to another specialized journal after making appropriate revisions.

Requested changes

1- "quasi-Hamiltonian" in Abstract might not be a standard terminology and could be replaced by a Floquet effective Hamiltonian as they use in the paper. 2- Is r in Eq.(2) defined? Is it $r\equiv|\mathbf{r}|$?

---

## Editorial Decision

awaiting_resubmission